# Genetic characteristics of *P. falciparum* parasites collected from 2012 to 2016 and anti-malaria resistance along the China-Myanmar border

**Mei Li**[1,2,3], **Hui Liu**[4]*, **Linhua Tang**[1,2,3], **Henglin Yang**[4], **Maria Dorina Geluz Bustos**[5], **Hong Tu**[1,2,3], **Pascal Ringwald**[6]

**1** National Institute of Parasitic Diseases, Chinese Center for Disease Control and Prevention (Chinese Center for Tropical Diseases Research), Beijing, China, **2** NHC Key Laboratory of Parasite and Vector Biology, WHO Collaborating Center for Tropical Diseases, Shanghai, China, **3** National Center for International Research on Tropical Diseases, Shanghai, 200025, China, **4** Yunnan Institute of Parasitic Diseases, Yunnan, 665000, China, **5** Office of the WHO Representative to Thailand, Bangkok, 10400, Thailand, **6** Coordinator Director Office, Global Malaria Programme, Geneva, Swizerland

* liubible@126.com

**Data Availability Statement:** All relevant data are within the paper and its Supporting Information files.

## Abstract

### Backgrounds

The therapeutic efficacy studies of DHA-PIP for uncomplicated *Plasmodium falciparum* patients were implemented from 2012 to 2016 along China (Yunnan province)-Myanmar border, which verified the high efficacy of DHA-PIP. With the samples collected in these studies, the genetic characteristics of *P. falciparum* parasites based on *in vivo* parasite clearance time (PCT) was investigated to explore if these parasites had developed resistance to DHA and PIP at molecular level.

### Methods

The genetic characteristics were investigated based on K13 genotypes, copy numbers of genes pfpm2 and pfmdr1, and nine microsatellite loci (Short Tandem Repeats, STR) flanking the K13 gene on chromosome 13. The PCT 50s were compared based on different K13 genotypes, sites, periods and copy numbers.

### Results

In the NW (North-West Yunnan province bordering with Myanmar) region, F446I was the main K13 genotype. No significant differences for PCT 50s presented among three K13 genotypes. In SW (South-West Yunnan province bordering with Myanmar) region, only wild K13 genotype was detected in all parasite isolates whose PCT 50s was significantly longer than those in NW region. For the copy numbers of genes, parasite isolates containing multiple copies of pfmdr1 gene were found in both regions, but only single copy of pfpm2 gene was detected. Though the prevalence of parasite isolates with multiple copies of pfmdr1 gene in SW region was higher than that in NW region, no difference in PCT 50s were

**Funding:** The recruitment, treatment of volunteers and samples collection was supported by the WHO Mekong Malaria Programme (WP/10/MVP/005837) from a Consolidated Grant from USAID-PMI to WHO; the Round 6 (CHN-607-G09-M) and 10 (CHN-011-G15-M) Global Fund grant to fight AIDS, Tuberculosis and Malaria (GFATM) in China. The study design, samples test, data collection and analysis, decision to publish and preparation of the manuscript was supported by the National Science Foundation of China (No. 81673113). We thank all participants for their contribution of time and patience in the study. We also thank staff of National Institute of Parasitic Diseases (NIPD) for logistic support, and clinical and laboratory staff of Menglian, Tengchong and Yingjiang County Center for Disease Control and Prevention for their hard work. We are also grateful for the support provided by the provincial health directors and the local health staff at the four sentinel sites. The opinions and recommendations expressed in this article are those of the authors and do not necessarily reflect the official views of WHO/USAID-PMI, GFATM, NIPD and YIPD.

**Competing interests:** The authors have declared that no competing interests exist.

**Abbreviations:** TES, Therapeutic Efficacy Studies; DHA-PIP, Dihydroartemisinin-Piperaquine; PCT, Parasite Clearance Time; STR, Short Tandem Repeats; NW area, North-West Yunnan province bordering with Myanmar; SW area, South -West Yunnan province bordering with Myanmar; W group, *P. falciparum* isolates with K13 wild genotype in NW area; Others group, *P. falciparum* isolates with Non-F446I K13 mutation; ML group, *P. falciparum* isolates from Menglian County in SW area; CQ, Chloroquine; ACT, Artemisinin-based Combination Therapy; GMS, Greater Mekong sub-region; SNP, Single Nucleotide Polymorphisms; WHO, World Health Organization.

presented between isolates with single and multiple copies of pfmdr1 gene. The median *He* values of F446I group and Others (Non-F446I K13 mutation) group were 0.08 and 0.41 respectively. The mean *He* values of ML group (Menglian County in SW) and W (wild K13 genotype in NW) group were 0 and 0.69 respectively. The mean F*st* values between ML and W groups were significantly higher than the other two K13 groups.

## Conclusions

*P. falciparum* isolates in NW and SW regions had very different genetic characteristics. The F446I was hypothesized to have independently appeared and spread in NW region from 2012 and 2016. The high susceptibility of PIP had ensured the efficacy of DHA-PIP *in vivo*. Multiple copy numbers of pfmdr1 gene might be a potential cause of prolonged clearance time of ACTs drugs along China-Myanmar border.

## Trial registration

Trial registration: ISRCTN, ISRCTN 11775446. Registered 17 April 2020—Retrospectively registered, the registered name was Investigating resistance to DHA-PIP for the treatment of Plasmodium falciparum malaria and chloroquine for the treatment of Plasmodium vivax malaria in Yunnan, China. http://www.isrctn.com/ISRCTN11775446.

## Backgrounds

Yunnan Province is located on the southwestern border of China, with 25 counties/cities bordering Myanmar, Laos and Vietnam. Suitable climate, unobstructed geographical environment and relatively low economic development, weak health and disease prevention capabilities make these areas the epidemic areas of multiple infectious diseases [1]. Malaria is one of the infectious diseases with quite severe incidence in these areas. During 2010–2016, from the launch of the National Malaria Elimination Action Plan in China to the last reported local case in China, both total and local cases in Yunnan province were the most among all provinces in China [2–8]. Most of them (78.02%, 4099/5254) were imported from Myanmar [9]. So, it is important to effectively and timely cure malaria patients there for achieving Malaria Elimination in China. However, the emergence and spread of malaria parasite resistance to anti-malaria drugs in these regions was a great challenge to this program.

Earlier *in vitro* assays detected a trend of declining sensitivity to artemisinins in the border area of Yunnan following a long history of unilateral application of artemisinin monotherapy since the 1970s [10–15]. Artemisinin-based Combination Therapy (ACT) was recommended for the treatment of uncomplicated *Plasmodium falciparum* malaria worldwide to counter the threat of resistance of *P. falciparum* to monotherapies [16–18]. In China, ACTs were used as the official first-line drugs to treat uncomplicated *P. falciparum* malaria starting in 2006. The recommended ACTs included Dihydroartemisinin-Piperaquine (DHA-PIP), artesunate-amodiaquine (AS-AQ), artimisinin-naphthoquine phosphate (ART-NQ), and artemisininpiperaquine (ART-PPQ). From 2007 to 2013, DHA-PIP sensitivity in *P. falciparum* had not significantly changed though an increasing trend of fever clearance time (FCT) and asexual parasite clearance times (APCT) presented in treating uncomplicated *P. falciparum* along China-Myanmar border [19]. Treatment with DHA-PIP initially demonstrated prolonged parasite clearance time (PCT), but began to fail as the partner drugs began failing in the GMS

around 2010 [20–33]. The decreasing efficiency of DHA-PIP here involved resistance to both DHA and PIP [20–30].

To monitor the resistance of *P. falciparum* parasites to ACTs, *in vivo*, *in vitro* methods and various molecular markers were developed and applied. Therapeutic efficacy studies (TES) along the China-Myanmar border, supported by WHO (World Health Organization), had been implemented since 2008 [19,34]. Overall, the clinical follow-ups indicated that DHA-PIP remained highly efficacious for treating uncomplicated *P. falciparum* between 2007–2013 [19]. However, the molecular data based on the *in vivo* study during 2012–2016 remained unknown, which were expected to explore the potential spread and development of malaria resistance. So, a retrospective genetic characterization of *P. falciparum* isolates collected in this period was carried out.

## Method

### Study sites

From 2012 to 2016, malaria patients were enrolled at four surveillance sites, Menglian county, Tengchong county, Yingjiang county, and Ruili city, all of which share a border with Myanmar. Among them, Menglian county belongs to the Pu'er prefecture located in the South-West (SW) Yunnan province. The other three sites are all part of the Baoshan and Dehong Prefecture, both of which are in the North-West (NW) Yunnan province.

### Study population

People who had traveled to Myanmar or had settled along the China-Myanmar border, and were infected with uncomplicated *P. falciparum* malaria, met the inclusion and exclusion criteria were recruited. However, unmarried women between 12 and 18 years of age were excluded. The qualifying participants that attended the study health clinic and aged between six months and 60 years old, were enrolled in the TES, treated on site with 3-day DHA-PIP, and were monitored weekly for 42 days. The follow-up was on a fixed schedule and consisted of check-up visits and along with clinical and laboratory examinations in accordance with the WHO protocol [35]. All adult patients signed an informed consent form for participation. Parents or guardians gave informed consent on behalf of their children. Children over 12 years of age signed the informed consent form.

### Treatment

*P. falciparum* patients were treated with DHA-PIP once daily for three consecutive days with a dose of 2 mg/kg/day DHA and 16 mg/kg/day PIP. One tablet of DHA-PIP contained 40 mg of DHA and 320 mg of PIP. DHA-PIP drugs were purchased from Chongqing Holley Healthpro Pharmaceutical CO., Ltd and supplied by the National Malaria Project of China. All doses of medicine were supervised, and patients were observed for adverse reactions for 30 min after medicine administration. The individual providing treatment signed the Case Report Form treatment sheet after every drug administration.

### Microscopic blood examination and blood collection

Thick and thin films of blood were made on slides on days D0, D1, D2, D3, D7, D14, D28, D35, D42. According to the WHO guidelines, treatment outcomes were classified based on an assessment of the parasitological and clinical outcome of antimalarial treatment [19]. All patients were classified as having early treatment failure, late clinical failure, late parasitological

failure, or an adequate clinical and parasitological response [35]. Two to three drops of blood were collected on filter paper on D0, D42, and on the day of failure.

## Calculating Parasite clearance time (PCTs)

Based on parasitemia data from D0 to D3 collected during the following-up of each case, PCTs for each case was calculated using the WHO parasite clearance estimator WHOApplication-6-18 [36]. Data of Slope of Half-life (HL) and duration when 50%, 75%, 90%, 95% and 99% of the parasites was cleared (PCT 50, PCT 75, PCT 90, PCT 95, and PCT 99) were collected. Only the PCT 50 data were compared among different genetic types and sites.

## Sequencing the K13-propeller domain

The K13-propeller domain was amplified according to the methods described by Ariey, et al [37]. The high-fidelity Taq DNA polymerase (Takara 9158A) was purchased from Takara Bio-medical Technology (Da Lian) Co., Ltd. The PCR products were sequenced by Map Biotech Co, Ltd. (Shanghai) and the sequences were aligned with that of wild Pf3D7 to confirm the single nucleotide polymorphisms (SNP). All parasite isolates with PCTs data were divided into different groups according to their K13 genotypes.

## Estimation of copy number pfpm2 and pfmdr1

DNA sequences of the pfpm2 and pfmdr1 genes were amplified and detected according to the methods described by Witkowski, et al [38]. The TB Green™ *Premix Ex Taq*™ II (Tli RNaseH Plus) used in the test was purchased from Takara Biomedical Technology (Da Lian) Co., Ltd. The copy numbers of pfpm2 were estimated by the following equation [38]:

$$Y = 0.4583x + 0.7109, (x = 2^{-\Delta Ct}, \Delta Ct = Ct_{pfpm2} - Ct_{Pf\beta-tubulin}).$$

The copy numbers of pfmdr1 were estimated by the following equitation [38]:

$$Y = 0.4497x + 0.8976, (x = 2^{-\Delta Ct}, \Delta Ct = Ct_{Pfmdr1} - Ct_{Pf\beta-tubulin}).$$

Samples with estimated copy numbers ≥1.56 (≈1.60) were defined as containing multiple copies, the others as single copy [38].

## Microsatellite loci genotyping

Nine microsatellite loci (−56.0Kb, −50Kb, −6.36Kb, 1.70Kb, −0.15Kb, 8.60Kb, 11Kb, 15.10Kb, 31.50Kb) flanking the K13 gene on chromosome 13 were tested according to the method described by Talundzic, et al. [16] and analyzed in Excel 2016 with method of GenAlEx 6501 [39]. Genetic diversity was estimated using expected heterozygosity (*He*) and coefficient of gene differentiation (F*st*) following the Chinese guideline (S1 File) introduced by Wang in 2016 [39].

## Data analyses

The data were calculated and compared in software SPSS 23. They were firstly treated with tests of normality (Shapiro-Wilk test). Values of PCTs and *He*s were presented in median and percentiles (25% Per and 75% Per), while F*st* values were in mean and 95% Confidence Interval (CI) according to different results of normality tests. Comparing methods of Kruskal Wallis H test, Mann-Whitney U test, Wilcoxon test and Fisher Exact test were applied in the data of

Non-normal distribution. Method of Paired T-test was applied in the data of normal distribution. Results with p-values of 0.05 or less were considered significant.

### Ethical approval

According to the Helsinki Declaration, ethical approval for the study was granted by the Ethics Committee of National Institute of Parasitic Diseases, China CDC and the Ethics Committee of the Yunnan Institute of Parasitic Diseases. The purpose of the study was explained and then approval was sought from patients and their caretakers. Informed written consent was obtained from patient or carers of Child patients. All results were kept confidential and were unlinked to any identifying information.

## Results

### General briefing of samples testing

In total, 198 recruited volunteers completed follow-ups between 2012 and 2016 and three were lost after D7 (partial data of their sample were used in the study). In the NW region, eight patients (5.00%, 8/160) were positive at D3 compared to one patient (2.44%,1/41) in the SW region. 174 isolates were successfully genotyped at K13, including 169 with PCTs data and 5 without. Of these, 134 were from the NW region and 40 were from the SW region. Among the 169 isolates, copy numbers of pfpm2 gene and pfmdr1 gene were obtained in 164 and 159 isolates, respectively. STR data from 139 isolates were successfully detected and further analyzed (Fig 1).

### Genotypes of K13 mutations

Among the 174 parasite isolates that were successfully genotyped at K13, 46.90% (99/174) presented with K13 mutations. All 99 isolates with positive K13 mutations were collected from the NW region with a total K13 mutation rate of 73.88% (99/134) and a F446I prevalence of 58.96% (79/134). Most of the isolates in the NW region were collected from Yingjiang county (59.20%, 103/134), which indicated a K13 mutation prevalence of 85.44% (88/103) (See Table 1). There were 40 isolates from the SW region with the K13 wild type.

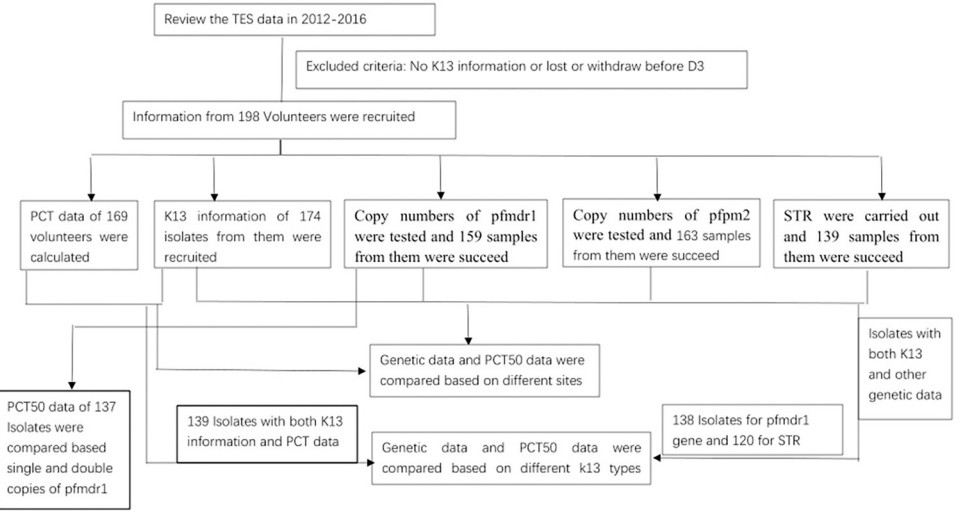

**Fig 1. Flow chart of sample test.**

**Table 1. Distribution of K13 mutations among different years and sites in the NW region.**

| K13 genotypes | 2012 | | 2013 | | 2014 | | 2015 | | 2016 | Sum |
|---|---|---|---|---|---|---|---|---|---|---|
| | Yingjiang | Tengchong | Ruili | Yingjiang | Yingjiang | Menglian | Yingjiang | Tengchong | Yingjiang | |
| C447S | 1 | 0 | 0 | 0 | 0 | 0 | 0 | 0 | 0 | 1 |
| F446I | 9 | 6 | 1 | 35 | 24 | 0 | 1 | 1 | 1 | 79 |
| N458Y | 1 | 1 | 0 | 0 | 0 | 0 | 0 | 0 | 0 | 2 |
| P574L | 2 | 0 | 0 | 4 | 1 | 0 | 0 | 0 | 0 | 7 |
| C447R | 1 | 0 | 0 | 0 | 0 | 0 | 0 | 0 | 0 | 1 |
| G495C | 1 | 0 | 0 | 0 | 0 | 0 | 0 | 0 | 0 | 1 |
| F483S | 1 | 0 | 0 | 0 | 0 | 0 | 0 | 0 | 0 | 1 |
| A676D | 1 | 0 | 1 | 0 | 1 | 0 | 0 | 0 | 0 | 2 |
| C580Y | 2 | 0 | 0 | 0 | 0 | 0 | 0 | 0 | 0 | 2 |
| C469Y | 0 | 0 | 0 | 0 | 2 | 0 | 0 | 0 | 0 | 2 |
| S466I | 0 | 0 | 1 | 0 | 0 | 0 | 0 | 0 | 0 | 1 |
| Wild | 10 | 18 | 2 | 0 | 2 | 40 | 0 | 0 | 3 | 75 |
| Total | 29 | 25 | 5 | 39 | 30 | 40 | 1 | 1 | 4 | 174 |

Note: This isolate showed double K13 mutations of N458Y and F446I. For the convenience of statistics and comparison, it recorded as N458Y.

A total of 11 K13 mutants were found, of which most of them were F446I (79.80%, 79/99), followed by P574L (7.07%, 7/99). The other mutants showed very low prevalence of 1.00–2.02%, such as C469Y (2.02%, 2/99), C580Y (2.02%, 2/99), N458Y (2.02%, 2/99). Most of the K13 mutants were found in Yingjiang County. In Tengchong county and Ruili city, only two K13 mutants were detected at each site (Table 1).

Based on the K13 mutants, 129 isolates with PCTs data in the NW region were divided into three groups. In detail, 34 isolates with the K13 wild type were classified as the W group, 75 with F446I as the F446I group, and 20 with other K13 mutational genotypes as the Others group. Forty isolates with the K13 wild genotype in Menglian county in the SW region were defined as the ML group.

## Comparison of PCT 50 based on K13 genotypes

The PCT data of four groups were shown in Table 2. The median values of HL, PCT50, PCT75, PCT9 and PCT99 in the NW region were 3.60, 12.89, 18.23, 23.56, 26.89 and 33.60 hours, respectively. The differences between different groups in NW region were not significant (Kruskal Wallis H test). The median values of HL and PCTs in the SW region (ML group) were 5.22, 15.81, 22.34, 28.75, 32.73 and 40.10 hours, respectively. The differences among 4 groups were significant (Kruskal Wallis H test). The difference presented both between ML and W group (Z = -3.09, -2.77,- 3.15, -3.20, -3.25, -3.28; P = 0.00, 0.01, 0.00, 0.00, 0.00, 0.00; Mann-Whitney U test), and between ML and F446I group (Z = -2.24, -2.17,- 2.24, -2.24, -2.22, -2.23; P = 0.03, 0.03, 0.03, 0.03, 0.03, 0.03; Mann-Whitney U test), but not between ML and Others group (Z = -1.04, -0.82,- 0.96, -0.86, -0.74, -0.71; P = 0.30, 0.42, 0.34, 0.39, 0.46, 0.48; Mann-Whitney test). However, median values of PCT 99 in all groups were less than 49 hours.

## Comparison of PCT 50 based on period for parasite isolates in the NW region

Details of K13 mutations of isolates collected in 2012 and between 2013 and 2014 were shown in Table 3. Among parasites collected in 2012, 48.15% (26/54) had K13 mutations, and 57.69%

**Table 2. PCTs data based on K13 genotypes.**

| Regions | Groups | Values | HL | PCT50 | PCT75 | PCT90 | PCT95 | PCT99 |
|---|---|---|---|---|---|---|---|---|
| **NW region** | W Group (n = 34) | Median values | 3.49 | 12.61 | 17.83 | 23.03 | 26.50 | 32.99 |
| | | 25% Percentile | 2.21 | 2.21 | 4.42 | 7.33 | 9.53 | 14.52 |
| | | 75% Percentile | 4.03 | 13.56 | 19.18 | 24.72 | 28.19 | 34.90 |
| | F446I Group (n = 75) | Median values | 3.55 | 12.89 | 18.23 | 23.58 | 27.11 | 33.60 |
| | | 25% Percentile | 3.01 | 11.85 | 16.75 | 21.59 | 24.63 | 30.53 |
| | | 75% Percentile | 4.95 | 14.16 | 20.06 | 26.56 | 30.86 | 38.73 |
| | Others Group (n = 20) | Median values | 4.64 | 13.29 | 19.07 | 24.99 | 28.84 | 36.60 |
| | | 25% Percentile | 2.04 | 2.04 | 4.07 | 6.75 | 8.78 | 13.42 |
| | | 75% Percentile | 8.45 | 21.18 | 30.59 | 40.48 | 47.06 | 58.08 |
| | **Subtotal** | Median values | 3.60 | 12.89 | 18.23 | 23.56 | 26.89 | 33.60 |
| | | 25% Percentile | 2.74 | 2.85 | 5.70 | 9.41 | 11.91 | 17.67 |
| | | 75% Percentile | 5.02 | 14.38 | 20.34 | 26.56 | 30.59 | 38.90 |
| | $\chi^2$ | | 1.79 | 2.35 | 2.46 | 2.68 | 2.89 | 3.24 |
| | df | | 2 | 2 | 2 | 2 | 2 | 2 |
| | P | | 0.41 | 0.31 | 0.29 | 0.26 | 0.24 | 0.20 |
| **SW region** | ML Group (n = 40) | Median values | 5.22 | 15.81 | 22.34 | 28.75 | 32.73 | 40.10 |
| | | 25Th Percentile | 2.10 | 2.10 | 4.20 | 6.98 | 9.07 | 13.86 |
| | | 75 Th Percentile | 9.71 | 21.83 | 30.71 | 39.39 | 44.81 | 54.41 |
| | $\chi^2$ | | 8.17 | 9.77 | 9.95 | 10.22 | 10.36 | 10.73 |
| | df | | 3 | 3 | 3 | 3 | 3 | 3 |
| | P | | 0.04 | 0.02 | 0.02 | 0.02 | 0.02 | 0.01 |

PCT: Parasites Clearance Time.

(15/26) of the mutations were F446I. The total F446I prevalence was 27.78% (15/54) in 2012. For isolates in 2013 and 2014 (2013–2014 isolate), 95.65% (66/69) of them had the K13 mutation and 86.36% (57/66) of the mutations were F446I. The total F446I prevalence was 82.61% (57/69) in this 2-year period, higher than that in 2012 (Fisher' exact test, P = 0.00). The comparison of median PCT 50 between isolates in 2012 (13.14, 25% Per = 2.33, 75% Per = 13.96, N = 54) and between 2013 and 2014 (12.89, 25% Per = 11.48, 75% Per = 15.63, N = 69) in the NW region demonstrated no difference (Z = -1.24, P = 0.21; Mann-Whitney U test).

## Comparison of PCT 50 based on sites in the NW region

The median PCT 50s of isolates in Tengchong and Yingjiang were 13.21 (25% Per = 6.76, 75% Per = 14.03, N = 28) and 12.87 ((25% Per = 2.88, 75% Per = 15.05, N = 102), respectively. No significant difference was present between these 2 surveillance sites (Z = -0.04, P = 0.97, Mann-Whitney test), but both lower than that in Menglian County (Z = -0.56, P = 0.01; Z = -2.51, P = 0.01;Mann-Whitney U test). Data from the Ruili site were not analyzed because only six cases were included.

**Table 3. K13 genotypes in different years for parasite isolates in NW region.**

| | Total No. of cases with PCT50 Data | No. of Cases with K13 mutations | No. of cases with F446I mutation | F446I prevalence (%) |
|---|---|---|---|---|
| **2012** | 54 | 26 | 15 | 27.78 |
| **2013–2014** | 69 | 66 | 57 | 82.61 |

## Amplification of pfpm2 and pfmdr1 gene

We did not observe any amplification of the pfpm2 gene in samples both in the NW region (N = 126) and in SW region (N = 16). Among the tested copy numbers of pfmdr1 gene from 159 samples, 14.63% (18/123) isolates from the NW region and 62.5% (10/16) from the SW region contained multiple copies (N = 16), respectively. The prevalence of isolates with multiple pfmdr1 copies in the SW region was significantly higher than that in the NW region (P = 0.00, Fisher exact test). The prevalence of isolates with multiple copies of pfmdr1 gene based on K13 genotypes were 19.35% (6/31), 13.04% (9/69) and 10% (2/20) in W group, F446I group and Others group, respectively. There was no difference between each 2 of them (Between W and F446I groups: P = 0.76; Between W and Others groups: P = 0.69; Between F446I and Others groups: P = 1.00; Fisher exact test). Totally, 13.48% (12/89) isolates presented both K13 mutations and multiple copies of pfmdr1 gene.

There was no significant difference in PCT 50s between isolates with single (Median = 13.17, 25% Per = 11.12, 75% Per = 15.32, N = 111) and multiple (Median = 13.48, 25% Per = 4.99, 75% Per = 19.33, N = 28) copies of pfmdr1 gene (Z = -0.76, P = 0.45). For the isolates with both K13 mutant and multiple copies, there was no significant difference observed in their PCT 50s (Median = 12.91, 25% Per = 2.81, 75% Per = 20.48, N = 12) with those presenting with single copy of pfmdr1 gene and wide K13 type (Median = 13.00, 25% Per = 2.16, 75% Per = 13.70, N = 25) (Z = -0.36, P = 0.72; Mann-Whitney U test).

## Reduced genetic *He* of the mutational K13 propeller allele and increased F*st* between parasites of ML group and W group

According to mean *He* values, the four groups were arranged as ML group, F446I group, Others group and W group from low to high (See Fig 2). In ML group from the SW region, *He* values of parasites at all loci sites was 0 while those in the NW region were all more than 0. The median *He* value of the F446I isolates at 9 microsatellite loci was 0.08 (25% Per = 0.03, 75% Per = 0.27, N = 9) with each individual value of less than 0.30 at all loci except of one that, which was 0.52 and located at upstream −56Kb of the K13 gene. The *He* values in the Others group fluctuated greatly from 0 to 0.65 with a median value of 0.41 (25% Per = 0.20, 75% Per = 0.61, N = 9). The mean *He* value in the W group was the highest among four groups, which was 0.69 (25% Per = 0.53, 75% Per = 0.78, N = 9). All the mean *He value*s showed significant difference with one another (Between W and F446I groups: Z = -2.67, P = 0.01; Between W and Others groups: Z = -2.67, P = 0.01; Between F446I and Others groups: Z = -2.10, P = 0.04; Wilcoxon test).

All F*st* values at nine different loci (See Table 4 [40]) between different groups were higher than or equal to 0.05. The mean F*st* values among four groups at nine different loci (0.41 ±0.07) were significantly higher than that among three NW groups (0.24±0.04) (Paired-T Test: t = −3.22, df = 8, P = 0.01). The highest difference between them presented at the sites of −0.15Kb (0.74 and 0.14) and 1.70Kb (0.47 and 0.21), which were closest to K13 genes among nine loci along chromosome 13. The mean F*st* values between the W group and each of other three groups at nine loci were also detailed in Table 4. The F*st* values between ML and W groups were significantly higher than other two pairs (Paired-T Test: t = − 2.66, df = 8, P = 0.03; t = −4.97, df = 8, P = 0.001), except that between the F446I and W groups at site of 1.70Kb to K13 gene show a higher value than that between the ML and W groups.

## Discussion

Drug resistant parasites tend to appear among neighboring countries where different drug policies are typically applied and parasite populations at borders would experience divergent drug

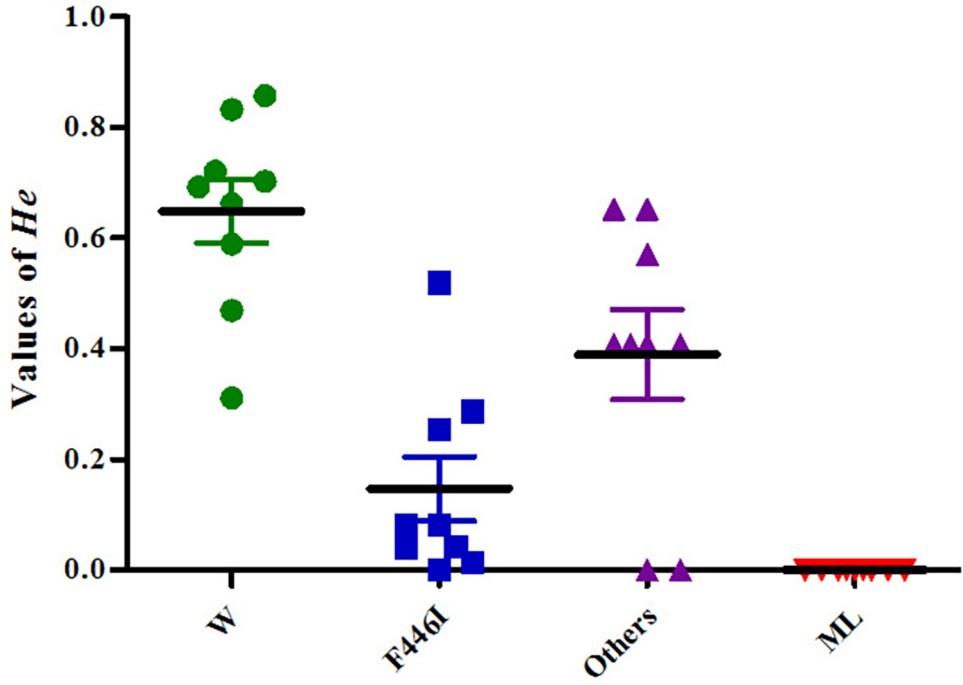

**Fig 2. *He* variation of different groups based on K13 genotypes.**

selection pressures [32,41–46]. In Yunnan province, Dehong, Baoshan and Pu'er Prefectures located at NW and SW region bordering Myanmar. During 2011–2019, imported malaria cases from Myanmar in these 3 prefectures accounted for 45.09% (2,369/5,254), 27.41%

**Table 4. *Fst* values at different loci and between different groups.**

| Loci | *Among* all 4 Groups | *Among* 3 NW Groups | Between F446I and W | Between Others and W | Between ML and W |
|---|---|---|---|---|---|
| −56.0Kb | 0.24 | 0.15 | 0.08 | 0.05 | 0.19 |
| −50Kb | 0.24 | 0.18 | 0.17 | 0.05 | 0.19 |
| −6.36Kb | 0.55 | 0.42 | 0.15 | 0.26 | 0.47 |
| −0.15Kb | 0.74 | 0.14 | 0.11 | 0.11 | 0.56 |
| 1.7Kb | 0.47 | 0.21 | 0.30 | 0.08 | 0.26 |
| 8.6Kb | 0.54 | 0.43 | 0.31 | 0.20 | 0.39 |
| 11Kb | 0.37 | 0.27 | 0.20 | 0.10 | 0.31 |
| 15.1Kb | 0.09 | 0.08 | 0.06 | 0.06 | 0.06 |
| 31.5Kb | 0.44 | 0.26 | 0.07 | 0.18 | 0.45 |
| **Mean values** | 0.41 [d] | 0.24 [ac] | 0.16 [bc] | 0.12 [b] | 0.32 [a] |
| **SE** | 0.07 | 0.04 | 0.03 | 0.03 | 0.05 |

a, b, c, d: If the same letter is present, no difference was found between the mean values according to paired T-Test; If the letters are different, significant differences were found between the mean values.

Differentiation Degree [40]: $0 \leq$ Fst $<0.05$, little genetic differentiation.

$0.05 \leq$ Fst $<0.15$, moderate differentiation.

$0.15 \leq$ Fst $<0.25$, great differentiation.

$0.25 \leq$ Fst, very great genetic differentiation.

(1,440/5,254) and 2.97% (156/5,254) of the total imported cases in Yunnan province, respectively. So, malaria patients were mainly enrolled in these Prefectures in those days.

Based on the SNPs examination and correction analysis using genomic tools, K13 mutants, such as C580Y, R539T, Y493H, and I543T, were verified to be correlated with the spread of resistance in west Cambodia and other GMS countries [37]. Consequently, F446I were identified as artemisinin partial resistance markers that affected only ring-stage parasites [47].

Previous molecular studies verified an outstanding genetic characteristic of *P. falciparum* isolates in the NW region, that F446I was the dominant K13 mutation allele, but no F446I isolates were found in the SW region [10,48–50]. Our findings in NW *P. falciparum* isolates supported these findings and revealed a wide spread of F446I in 2013 and 2014 following with civil war in Kachin, Myanmar based on the accumulation in 2012 or/and earlier. We presumed the F446I in NW region was a recent drug-mediated mutation due to ACTs used in the region to a new arrival since no F446I had been reported in the SW region. This genetic mutation was distinctly different from other GMS countries, such as in Cambodia and Thailand where the dominant K13 mutation was C580Y, though F446I allele was sporadically reported there. This difference might be due to the parasites' accumulative adaptations to their sustained different host environments [51]. In addition, between 2014 and 2016, a new mutation (G533S), which showed significantly higher ring survival rates than the wide type appeared and reached a prevalence of 44.1% along the China-Myanmar border [52]. The K13 mutant R561H, which was confirmed as driving artemisinin resistance, was identified in 19 of 257 (7.4%) patients at Masaka in Rwandan and phylogenetic analysis revealed they were indigenous lineage [53]. Based on this evidence, it seemed that with the exception of long-distance spreading because of migrant population, new mutations related to artemisinini resistance often appear independently among *P. falciparum* isolates at the locations where they received accumulative pressures from different human populations, anti-malaria medicines, and even vectors, and then spread around the hot points. In our study, the mean F$st$ value of 0.161 (±0.032) at nine microsatellite loci suggests that F446I had significantly differentiated genetically from the W group. However, the lowest mean *He* value (0.15, 0.014–0.28, N = 71) was observed in F446I isolates. This value was also lower than that of C580Y (0.35±0.08) in Thailand [16]. Low *He* values meant low diversity within these isolates, thus indicating that F446I isolates had experienced short selection pressure or little gene exchanges with other genotypes since they emerged. This result gave more evidence to the hypothesis that, F446I isolates have independently emerged in those years in the process of adapting to its environmental pressures, which then spread along the NW Yunnan-Myanmar border. This inference agreed with the consecutive and dramatic increase of prevalence of K13 mutations in different years in this region [50]. With the global application of artemisnin, this pattern of interaction between *P. falciparum* parasites and artemsinin drugs warranted more attention.

In this study, the high prevalence of mutant K13 was mainly due to F446I, suggested that PIP had undergone more pressure in the NW region during 2012–2016, though no great influence to the therapeutic efficacy of DHA-PIP *in vivo* appeared, which was due to the high susceptibility of PIP [19,54]. Therefore, monitoring the efficacy of antimalarial drugs *in vivo*, using molecular surveillance of both artemisinin and long-acting partner drugs, must be sustained. However, this study was not planned for PCT (6 hr smear preparation and examination), so that it perhaps was unable to provide the actual data to the association between PCT and mutations. Thus, no significantly different PCT 50s between isolates of the F446I and W groups was observed, which was various from other reports [50].

The pfmdr1 genotype is correlated with resistance of *P. falciparum* to Chloroquine (CQ), Mefloquine (MQ), and artemisinins, whereas knockdown of pfmdr1 expression leads to increased susceptibility [55]. In the 1990s, a series of *in vitro* assays demonstrated that

resistance to multiple antimalarial drugs was present in the South and West Yunnan Province bordering with Myanmar, which included both PIP and some artemisinin drugs [11–14]. In our study, the multiple copies appeared in both NW and SW regions. These findings were different from other reports before 2012 in the NW region [56,57] in which only isolates with single pfmdr1 copy detected. They were also different from the reports in eastern Cambodia where DHA-PIP treatment failures increased steadily in 2014 to reach a high frequency by 2015[38]. According to that report, most treatment failures had a single gene copy of mdr1 (112 [94·1%] of 119), confirming earlier reports of failures [38]. So artesunate plus mefloquine was suggested a viable option to treat DHA-PIP failures in Cambodia. Obviously, this suggestion was not applicable in China-Myanmar border, though MQ had never been deployed in the NW region, and neither *in vivo* or *in vitro* MQ resistance had been detected in the SW region before 2000 [58–61]. Quite a few studies revealed that parasites with increased pfmdr1 copies would significantly reduce *in vitro* susceptibility to artemisinin derivatives, such as artesunate, which affected the treatment effect seriously in the GMS [62–66]. The emergence of pfmdr1 amplification trend in China-Myanmar border might imply falciparum isolates had adopted this mechanism to cope with stress from artemisinin (DHA) besides of K13 mutations in the NW region. This adaption was also different from the F32-ART5 lineage which undergoing 5-year selection of artemisinin derivatives but showed negative pfmdr1 amplification [37]. This once again proves the diversity of ways in which falciparum malaria parasites fight against artemisinin. Though isolates with both K13 mutant and pfmdr1 amplification were observed in the NW region, no synergistic effect on resistance to atimisinin was observed between them.

In the SW region, though no K13 mutants detected, the higher PCTs and prevalence of isolates with multiple copies of pfmdr1 gene might indicate they resisted artemisinin (DHA) by this mechanism and achieved success within a certain range. Unfortunately, no correlation between PCT 50 values and copy numbers of the pfmdr1 gene was found. So, more studies, such as *in vitro* assays of these isolates based on single drug were placed in high hopes to be able to reveal the truth. Multiple copy numbers of pfmdr1 genes should be considered as one of potential causes of decreased prolonged clearance time of *P. falciparum* parasites along China-Myanmar border [50,67]. Thus, testing the copy number of pfmdr1 gene can be used as a molecular marker to monitor the susceptibility of artemisinin derivatives to imported *P. falciparum* and as part of a surveillance network of anti-malarial drugs in China [54].

Because the resistance of *P. falciparum* to the fast-acting artemtisinin component means longer exposure of the parasite to the long-acting partner drugs, there is an increased probability of parasite survival [20–30]. Genome-wide associated studies revealed that amplification of two protease genes, plasmepsinII–III (pfpm2/3), were associated with clinical resistance to PIP in Cambodia. Based on this achievement, a test system with the pfpm2 gene, as a marker for resistance of parasites to PIP, was established [38,68]. The negative results in our study ensured the efficacy of DHA-PIP *in vivo*. STR data also revealed that isolates in the ML group showed that there was low diversity within the group but had a higher genetic differentiation from the other groups. The former was consistent with their background that a malaria outbreak occurred in that region in 2014 [69]. The latter indicated the genetic characteristics of *P. falciparum* isolates in the NW and SW regions diverged remarkably. This result supported the point that isolates in SW have applied a different ways to resist the pressure from artemisinin.

## Conclusions

According to our study, *P. falciparum* isolates in the NW and SW regions have very different genetic characteristics. Artemisinin partial resistance, inferred from the genetic marker F446I,

had independently appeared and spread in the NW region during 2012 and 2016. With the global application of ACTs, this pattern of emerging artemisinin resistance is worth further investigation. PIP-resistant featured markers, based on genetic analysis (pfpm2), showed negative results, which ensured the efficacy of DHA-PIP *in vivo*. In addition to monitoring the efficacy of antimalarial drugs *in vivo*, molecular surveillance of both artemisinin and long-acting partner drugs was strongly advised to be sustained. To monitor susceptibility of artemisinin derivatives to *P. falciparum* parasites, with the exception of the K13 mutation, the application of the copy number of the pfmdr1 gene was recommended in the anti-malarial drug surveillance network in China.

## Limitations

In this study, HL and PCT values were calculated using the WHO parasite clearance estimator, WHO Application, based on a schedule of 24-hourly sampling, which was not as accurate as that of 6 or 8-hourly sampling. Thus, a comparison of HL or PCT values with data in other studies could not be made, and association between PCT and mutation may not provide the actual data. Gene amplification is more frequent than point mutation in *P. falciparum* parasites. Mdr1 mutation analysis might give more information about the different PCTs between NW and SW region. However, this study was not carried out because of limitation of sample quantity. To obtain accurate copy number of pfpm2 and pfmdr1 gene, amplification efficiency for DNA fragments detection of pfpm2, pfmdr1and pfβ-tubulin gene, and new equation curve between copy number and $2^{-\Delta t}$ values based on 90–110% amplification efficiency are recommended in new laboratory before sample test. Negative control (Pf 3D7) should be included in each run that might clear the difference of Ct values between different genes as being pointed [38,70].

## Supporting information

**S1 File. Guideline for calculating H*e* and F*st* (In Chinese).** Photos of relevant content in reference 39.
(PDF)

## Acknowledgments

We thank all participants for their contribution of time and patience in the study. We also thank staff of National Institute of Parasitic Diseases (NIPD) for logistic support, and clinical and laboratory staff of Menglian, Tengchong, Ruili and Yingjiang County Center for Disease Control and Prevention for their hard work. We are also grateful for the support provided by the provincial health directors and the local health staff at the four sentinel sites. The opinions and recommendations expressed in this article are those of the authors and do not necessarily reflect the official decisions, policy, or views of WHO, USAID-PMI, GFATM, NIPD and YIPD.

We also thank LetPub (www.letpub.com) for linguistic assistance and pre-submission expert review.

## Author Contributions

**Conceptualization:** Mei Li.

**Data curation:** Mei Li, Hui Liu.

**Formal analysis:** Mei Li, Hong Tu.

**Funding acquisition:** Hui Liu, Linhua Tang.

**Investigation:** Mei Li, Hui Liu, Linhua Tang, Henglin Yang.

**Methodology:** Mei Li, Maria Dorina Geluz Bustos.

**Project administration:** Mei Li, Hui Liu, Henglin Yang.

**Resources:** Mei Li, Hui Liu.

**Software:** Maria Dorina Geluz Bustos.

**Supervision:** Mei Li, Linhua Tang, Henglin Yang, Maria Dorina Geluz Bustos.

**Writing – original draft:** Mei Li.

**Writing – review & editing:** Mei Li, Maria Dorina Geluz Bustos, Pascal Ringwald.

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
