## [Decision Letter · Decision Letter 0]

15 Mar 2023

PONE-D-23-03162

Genetic characteristics of P. falciparum parasites collected from 2012 to 2016 and anti-malaria resistance along the China-Myanmar border

PLOS ONE

Dear Dr. Li,

Thank you for submitting your manuscript to PLOS ONE. After careful consideration, we feel that it has merit but does not fully meet PLOS ONE’s publication criteria as it currently stands. Therefore, we invite you to submit a revised version of the manuscript that addresses the points raised during the review process.

We encourage you to submit your revision by Apr 28 2023 11:59PM. If you will need more time than this to complete your revisions, please reply to this message or contact the journal office at plosone@plos.org. Please include the following items when submitting your revised manuscript:A rebuttal letter that responds to each point raised by the academic editor and reviewer(s). You should upload this letter as a separate file labeled 'Response to Reviewers'.A marked-up copy of your manuscript that highlights changes made to the original version. You should upload this as a separate file labeled 'Revised Manuscript with Track Changes'.An unmarked version of your revised paper without tracked changes. You should upload this as a separate file labeled 'Manuscript'.

We look forward to receiving your revised manuscript.

Kind regards,

Himanshu Gupta

Academic Editor

PLOS ONE

   " WHO Mekong Malaria Programme (WP/10/MVP/005837) from a Consolidated Grant from USAID-PMI to WHO; the Round 6 (CHN-607-G09-M) and 10 (CHN-011-G15-M) Global Fund grant to fight AIDS, Tuberculosis and Malaria (GFATM) in China. It was also supported by the National Science Foundation of China (No. 81673113)" 

7. We note that Figure 1 in your submission contain [map/satellite] images which may be copyrighted. All PLOS content is published under the Creative Commons Attribution License (CC BY 4.0), which means that the manuscript, images, and Supporting Information files will be freely available online, and any third party is permitted to access, download, copy, distribute, and use these materials in any way, even commercially, with proper attribution. For these reasons, we cannot publish previously copyrighted maps or satellite images created using proprietary data, such as Google software (Google Maps, Street View, and Earth). For more information, see our copyright guidelines: http://journals.plos.org/plosone/s/licenses-and-copyright.

Reviewers' comments:

Reviewer's Responses to Questions

**Comments to the Author**

1. Is the manuscript technically sound, and do the data support the conclusions?

Reviewer #1: Yes

Reviewer #2: Yes

2. Has the statistical analysis been performed appropriately and rigorously? 

Reviewer #1: Yes

Reviewer #2: I Don't Know

3. Have the authors made all data underlying the findings in their manuscript fully available?

Reviewer #1: No

Reviewer #2: Yes

4. Is the manuscript presented in an intelligible fashion and written in standard English?

Reviewer #1: No

Reviewer #2: Yes

5. Review Comments to the Author

Reviewer #1: Overview:

Li et al’s manuscript titled Genetic characteristics of P. falciparum parasites collected from 2012 to 2016 and anti-malaria resistance along the China-Myanmar border examined the frequencies of several molecular markers associated with delayed artemisinin clearance and measured clearance times of P. falciparum strains collected along the China-Myanmar border. This study is interesting, as it takes place in the decade immediately preceding China’s malaria-free certification in 2021.

The primary purpose of the study was to assess the level of delayed artemisinin clearance at the China-Myanmar border. To do this, they assessed SNPs in the K13-propellar domain microsatellites flanking the K13 gene on chromosome 13, and copy number variation of pfmdr1 and pfpm2. Clearance times following DHA-PIP treatment were also measured.

Overall, the actual results appear to be sound, and I recommend publication. The observation of different genetic structure in the NW and SW collection sites is interesting and appears to be consistent with the fragmented parasite populations seen in SE Asia.

However, the language style will need to be revised before final acceptance. The manuscript is understandable but awkward in too many areas to list. I have, however, left some recommendations specific recommendations to improve the presentation of the paper.

Major comments

Introduction

1) The introduction should be refocused to provide greater epidemiological context to the importance of the China-Myanmar border. It is obvious that a major motivation for the study was to study the drug resistance profiles of individuals moving between China and Myanmar, which was important because China was actively applying for malaria-free certification in this time period

a. The introduction can be improved by explaining the status of malaria transmission in Yunnan province between 2012-2016, and the drug resistance profile in Myanmar (or possibly Laos), as these are likely where the parasites seen at the China-Myanmar border are coming from.

2) Too much time is spent describing CQ resistance and its spread. --CQ is not relevant to this study and I would advise focusing specifically on describing what is known about ACTs and DHA-PIP. For example, much of lines 68-75 can be cut.

3) Line 61 – “In response to widespread resistance….” is technically correct but skips a lot of steps. ACTs were used after many drugs, including CQ began to fail due to the emergence of drug resistance.

Methods

1) The map used in Fig 1 should be rescaled so that it is zoomed into the China-Myanmar border. The actual study sites are too small to see.

2) Line 179 – “all data were collected in Excel 2016” – I don’t think this is relevant detail

3) Line 181 – “Column statistics Method in Column Analysis” is uninformative and the name/description of the exact test being used should be used instead

Results

1) The description of the sampling design could be improved by including a flow chart or some other visualization to help the reader understand and remember

2) Line 195 – The term “K13 genotype” is misleading—I think it can be clarified to state “Samples that were successfully genotyped at K13”. Same at Line 200 – It should be rephrased as “Among the 175 parasites that were successfully genotyped at K13…”

3) Table 1 is confusing because it is unclear which regions correspond to which year. Example: was Ruili collected in 2012 or 2013? Some shading or coloring would help clarify

4) Were any parasites observed with multiple K13 mutations?

5) Line 262 replace with “We did not observe any amplification of the pfpm2 gene in our samples”.

6) Line 263 – It is strange that the estimated CNV for pfpm2 is less than 1, as this would imply a deletion in some of the samples. I am guessing this is because the equations used to estimate CNV are not completely accurate—but if there truly are deletions of pfpm2 in the data that would be useful to report

a. Line 263 – “NW region significantly lower…” – this might be technically true, but I doubt that this is due to actual biology… I think it is more likely to be because the equation used is not completely accurate

7) Line 278: the name of the test should be used (I think it is an ANOVA?)

Discussion

1) Line 325-326 – “Following the spread of chloroquine….”, Technically true, but ultimately irrelevant. Would suggest this first section focus more on the specific implications of drug resistance of at the China-Myanmar border. Lines 329-331 should be expanded upon

2) The most interesting finding to me is that PCTs in the W group are much lower than the PCT values in the ML group. Both groups are wild type at K13, which would imply that the ML group has other mutations contributing to delayed clearance—I think this is an important finding and should be discussed more

3) Lines 347-350 – I think the phrasing can be improved. The low He observed in F466I could be due to strong, recent drug-mediated positive selection for the mutation due to ACT use in the region. It could also indicate that F466I arrived at in Yunnan border from a recent importation event that rapidly expanded in NW Yunnan province – both could result in extremely low He.

4) Lines 353-357 – This is also interesting and should be expanded. Given that the samples in SW Yunnan were collected during an outbreak, it is entirely possible that the parasites in the ML group are all clonal or extremely clonal to one another and derived from a recent importation event.

5) It is curious that C580Y was not found at high frequencies in this study. Some discussion about why this might be would be useful. Additional discussion for why F446I is so prevalent would also be interesting

Minor Comments

1) In general, the name of the statistical test used to estimate p-values should be included when reporting p-values

2) Line 160 – Pf3D7 is misspelled (Pf37D)

3) lines 167 and 169-Equation is misspelled (“equitation”) in

4) Line 176-177: The equations used for He and Fst could be useful

5) The results from SW region should also be included in Table 1, even though there were no K13 mutants observed. Alternatively, Table 1 should be renamed “Distribution of K13 genotypes in NW Yunnan”

6) Line 268 – the section starting with “The prevalence of isolates with multiple pfmdr1 copies…” should be made into its own paragraph

7) The language style of the manuscript should be reviewed by a dedicated editor

Reviewer #2: 1. It would be worth mentioning about the drug policy in the different study area. Although author mentioned the drug of choice in line no. 98-101 but if possible then mentioned the specific drug used in the study area.

2. Since the study follows the standard WHO protocol for efficacy monitoring, the sample size as per WHO protocol was not adequate to conclude for each site vs each year which need to explain.

3. The study was not planned for PCT (6 hr smear preparation and examine) so association between PCT and mutation may not provide the actual data. Therefore, author needs to clarify in starting while discussing the associations.

4. Author did analyse only the copy number variation in MDR1 gene and explained on the association of SW region vs NW region however, author did not perform the mdr1 mutation analysis so that the statement may be more conclusive as mutation along with CNV play a role in artemisinin resistance. If not possible then it should come into the study limitation.

5. F446I mutation are found particularly one study sites (Yingjiang) with lower frequency in 2012 (35%) and subsequently it raised to 90% in 2013 and 2014. Please discuss the possible cause of the spread particularly in one site. If possible, the corelate the epidemiological data of the study site for recombination point of view.

6. What were the QA/QC procedures for K13 sequencing? Were the sequences of the F446I containing samples confirmed; either by re-sequencing, or any other method?

6. PLOS authors have the option to publish the peer review history of their article (what does this mean?). If published, this will include your full peer review and any attached files.

Reviewer #1: No

Reviewer #2: **Yes: **Praveen K Bharti

---

## [Author Response · Author response to Decision Letter 0]

11 Jul 2023

Response to Reviewers

1. Response to academic editor

1.1 Please ensure that your manuscript meets PLOS ONE's style requirements, including those for file naming. The PLOS ONE style templates can be found at 

Response: The style of the manuscript has revised according to the templates. 

1.2 We note that the grant information you provided in the ‘Funding Information’ and ‘Financial Disclosure’ sections do not match. When you resubmit, please ensure that you provide the correct grant numbers for the awards you received for your study in the ‘Funding Information’ section.

Response: Have done as your advice

1.3 Thank you for stating the following financial disclosure: 

 " WHO Mekong Malaria Programme (WP/10/MVP/005837) from a Consolidated Grant from USAID-PMI to WHO; the Round 6 (CHN-607-G09-M) and 10 (CHN-011-G15-M) Global Fund grant to fight AIDS, Tuberculosis and Malaria (GFATM) in China. It was also supported by the National Science Foundation of China (No. 81673113)" Please state what role the funders took in the study. If the funders had no role, please state: "The funders had no role in study design, data collection and analysis, decision to publish, or preparation of the manuscript." 

Response：The recruitment of volunteers, diagnosis, treatment, samples collection and molecular test (K13 gene) were supported by WHO Mekong Malaria Programme. Molecular tests (other genes), expert review and publication supported by National Science Foundation of China.

1.4 We note that you have stated that you will provide repository information for your data at acceptance. Should your manuscript be accepted for publication, we will hold it until you provide the relevant accession numbers or DOIs necessary to access your data. If you wish to make changes to your Data Availability statement, please describe these changes in your cover letter and we will update your Data Availability statement to reflect the information you provide.

Response： Supply as appendix 1

1.5 PLOS requires an ORCID iD for the corresponding author in Editorial Manager on papers submitted after December 6th, 2016. Please ensure that you have an ORCID iD and that it is validated in Editorial Manager. To do this, go to ‘Update my Information’ (in the upper left-hand corner of the main menu), and click on the Fetch/Validate link next to the ORCID field. This will take you to the ORCID site and allow you to create a new iD or authenticate a pre-existing iD in Editorial Manager. Please see the following video for instructions on linking an ORCID iD to your Editorial Manager account: https://www.youtube.com/watch?v=_xcclfuvtxQ

Response: Have done

1.6 Please include your full ethics statement in the ‘Methods’ section of your manuscript file. In your statement, please include the full name of the IRB or ethics committee who approved or waived your study, as well as whether or not you obtained informed written or verbal consent. If consent was waived for your study, please include this information in your statement as well. 

Response: Yes, all required information had been included in the ‘Methods’ section. 

1.7 We note that Figure 1 in your submission contain [map/satellite] images which may be copyrighted. All PLOS content is published under the Creative Commons Attribution License (CC BY 4.0), which means that the manuscript, images, and Supporting Information files will be freely available online, and any third party is permitted to access, download, copy, distribute, and use these materials in any way, even commercially, with proper attribution. For these reasons, we cannot publish previously copyrighted maps or satellite images created using proprietary data, such as Google software (Google Maps, Street View, and Earth). For more information, see our copyright guidelines: http://journals.plos.org/plosone/s/licenses-and-copyright.

Response: The figure was deleted. The information of surveillance sites were supplied alongside K13 sequences (Appendix 1). 

1.8 Please include captions for your Supporting Information files at the end of your manuscript, and update any in-text citations to match accordingly. Please see our Supporting Information guidelines for more information: http://journals.plos.org/plosone/s/supporting-information. 

Response: Have done as guidelines.

2. Response to Reviewers

2.1 Response to Reviewer1

2.1.1 Intoduction

1) The introduction should be refocused to provide greater epidemiological context to the importance of the China-Myanmar border. It is obvious that a major motivation for the study was to study the drug resistance profiles of individuals moving between China and Myanmar, which was important because China was actively applying for malaria-free certification in this time period

Response: The introduction (background) have revised according to reviewer’s advice.

a. The introduction can be improved by explaining the status of malaria transmission in Yunnan province between 2012-2016, and the drug resistance profile in Myanmar (or possibly Laos), as these are likely where the parasites seen at the China-Myanmar border are coming from.

Response: The introduction (background) have revised according to reviewer’s advice.

2) Too much time is spent describing CQ resistance and its spread. --CQ is not relevant to this study and I would advise focusing specifically on describing what is known about ACTs and DHA-PIP. For example, much of lines 68-75 can be cut.

Response: The introduction (background) have revised according to reviewer’s advice.

3) Line 61 – “In response to widespread resistance….” is technically correct but skips a lot of steps. ACTs were used after many drugs, including CQ began to fail due to the emergence of drug resistance.

Response: The introduction (background) have revised according to reviewer’s advice.

2.1.2 Methods

1) The map used in Fig 1 should be rescaled so that it is zoomed into the China-Myanmar border. The actual study sites are too small to see.

Response: The figure was deleted because of copyright. The information of surveillance sites were supplied alongside K13 sequences (Appendix 1).

2) Line 179 – “all data were collected in Excel 2016” – I don’t think this is relevant detail

Response: This sentence has been deleted. 

3) Line 181 – “Column statistics Method in Column Analysis” is uninformative and the name/description of the exact test being used should be used instead

Response: The statistics methods were re-considered. The data were re-analyzed in SPSS 23. So some data in Results section are different from previous version. For example, PCT values were presented in median and percentiles (25% Per and 75% Per) replacing of mean and CI, because the data was non-normal distributed. All these changes had not influenced the conclusion. 

2.1.3 Results

1) The description of the sampling design could be improved by including a flow chart or some other visualization to help the reader understand and remember

Response: Have added “CONSORT flowchart” as Fig.1 to describe this part. Previous Fig 2, Fig 3 and Fig 4 were deleted, because their information had included in tables. 

2) Line 195 – The term “K13 genotype” is misleading—I think it can be clarified to state “Samples that were successfully genotyped at K13”. Same at Line 200 – It should be rephrased as “Among the 175 parasites that were successfully genotyped at K13…”

Response: Have revised according to reviewer’s advice.

3) Table 1 is confusing because it is unclear which regions correspond to which year. Example: was Ruili collected in 2012 or 2013? Some shading or coloring would help clarify

Response: Have revised according to reviewer’s advice. Samples in Ruili were collected in 2013. 

4) Were any parasites observed with multiple K13 mutations?

Response: Yes, one multiple K13 mutations N458Y+F446I was found. However, it was record as N458Y in this manuscript because N458Y was verified to be connected with resistance of falciparum parasite to artimisinin, also for the convenience of comparative analysis.

5) Line 262 replace with “We did not observe any amplification of the pfpm2 gene in our samples”.

Response: Have revised according to reviewer’s advice.

6) Line 263 – It is strange that the estimated CNV for pfpm2 is less than 1, as this would imply a deletion in some of the samples. I am guessing this is because the equations used to estimate CNV are not completely accurate—but if there truly are deletions of pfpm2 in the data that would be useful to report

Response: Yes, we agree with reviewer’s view that CNV are not completely accurate because we deduce them from the equation curve in other report directly. But we are unable to ensure if there are deletions because of our shortage in more professional knowledge to explain it. To ensure accurate judgement, we re-check and re-calculate the original data. We draw a conclusion that CNV might be not completely accurate, but the judgement of single or multiple copies was right. So, the results were compared based on positive or negative amplification replacing of numerical value and the values of the copies were not shown. 

Here was the supporting information of reference sample Pf3D7

The PCR results (Ct Values) of Pf 3D7 isolate

 Pfmdr1 PfPM2 Pfβ-tubulin

Repeat 1 17.42 17.08 17.20

Repeat 1 17.15 16.64 17.03

Repeat 1 17.15 17.09 17.09

Average 17.24 ±0.16(CV 0.90%) 16.94±0.26 (CV 1.52%) 17.11±0.09 (CV 0.50%)

a. Line 263 – “NW region significantly lower…” – this might be technically true, but I doubt that this is due to actual biology… I think it is more likely to be because the equation used is not completely accurate

Response: Since the data is not accurate, we use positive and negative amplification to compare the difference in different region or groups.

7) Line 278: the name of the test should be used (I think it is an ANOVA?)

Response: Have revised according to reviewer’s advice. However, it is not ANOVA but Kruskal Wallis H test applied in comparing multiple groups of independent samples when their data non-normally distributed.

2.1.4 Discussion

1) Line 325-326 – “Following the spread of chloroquine….”, Technically true, but ultimately irrelevant. Would suggest this first section focus more on the specific implications of drug resistance of at the China-Myanmar border. Lines 329-331 should be expanded upon

Response: Have revised according to reviewer’s advice. Most of the statement about CQ was deleted in this manuscript.

2) The most interesting finding to me is that PCTs in the W group are much lower than the PCT values in the ML group. Both groups are wild type at K13, which would imply that the ML group has other mutations contributing to delayed clearance—I think this is an important finding and should be discussed more.

Response: I have tried my best to revise the discussion according to reviewer’s advice. I am not sure if it is full.

3) Lines 347-350 – I think the phrasing can be improved. The low He observed in F466I could be due to strong, recent drug-mediated positive selection for the mutation due to ACT use in the region. It could also indicate that F466I arrived at in Yunnan border from a recent importation event that rapidly expanded in NW Yunnan province – both could result in extremely low He.

Response: Have revised according to reviewer’s advice. For the F446I has not detected in South-West Yunnan province. We prefer recent drug-mediated positive selection to new arrival.

4) Lines 353-357 – This is also interesting and should be expanded. Given that the samples in SW Yunnan were collected during an outbreak, it is entirely possible that the parasites in the ML group are all clonal or extremely clonal to one another and derived from a recent importation event.

Response: Have revised according to reviewer’s advice. 

5) It is curious that C580Y was not found at high frequencies in this study. Some discussion about why this might be would be useful. Additional discussion for why F446I is so prevalent would also be interesting

Response: In fact, I could not ensure the reason. However, according to the C580Y in Cambodia, G533S in NW region in 2014-2016, F446I in NW region in 2012-2016, and R561H in Africa, we include that this might the character of falciparum parasites to artimisinin in different place with different background. 

2.1.5 Minor Comments

1) In general, the name of the statistical test used to estimate p-values should be ncluded when reporting p-values

Response: Have revised according to reviewer’s advice.

2) Line 160 – Pf3D7 is misspelled (Pf37D)

Response: Have revised according to reviewer’s advice.

3) lines 167 and 169-Equation is misspelled (“equitation”) in

Response: Have revised according to reviewer’s advice.

4) Line 176-177: The equations used for He and Fst could be useful

Response: Yes, we agree with reviewer’s opinion that equations could be useful for other researchers. So we recheck the reference book “Wang ZF ed. Molecular ecology and fundamentals of data analysis. Science Press. Beijing:2016.” Unfortunately, this book was focused on the software operation process, not the theory. No concerned equations were found. And none of the partners of this study were good at analysis of genetic data. So, here could not supply the equations. If it was accessible to us one day, we are willing to share it. 

5) The results from SW region should also be included in Table 1, even though there were no K13 mutants observed. Alternatively, Table 1 should be renamed “Distribution of K13 genotypes in NW Yunnan”

Response: Have revised to “Distribution of K13 genotypes in the NW region” according to reviewer’s advice.

6) Line 268 – the section starting with “The prevalence of isolates with multiple pfmdr1 copies…” should be made into its own paragraph

Response: Have revised according to reviewer’s advice.

7) The language style of the manuscript should be reviewed by a dedicated editor

2.2 Response to Reviewer 2: 

2.2.1 It would be worth mentioning about the drug policy in the different study area. Although author mentioned the drug of choice in line no. 98-101 but if possible then mentioned the specific drug used in the study area.

Response: Have revised according to reviewer’s advice.

2.2.2 Since the study follows the standard WHO protocol for efficacy monitoring, the sample size as per WHO protocol was not adequate to conclude for each site vs each year which need to explain.

Response: Yes, the sample size was not adequate, because of the decreased incidence of malaria cases in this area. However, we have enroll as many as we can. Otherwise, because this study was focus on the genetic characteristics of Pf in this area, not a report of the TES results, no explain for the sample size presented in this manuscript. Perhaps, our partner would give some interpret in a new manuscript.

2.2.3 The study was not planned for PCT (6 hr smear preparation and examine) so association between PCT and mutation may not provide the actual data. Therefore, author needs to clarify in starting while discussing the associations.

Response: Have revised according to reviewer’s advice.

2.2.4 Author did analyse only the copy number variation in MDR1 gene and explained on the association of SW region vs NW region however, author did not perform the mdr1 mutation analysis so that the statement may be more conclusive as mutation along with CNV play a role in artemisinin resistance. If not possible then it should come into the study limitation.

Response: Have revised according to reviewer’s advice.

2.2.5 F446I mutation are found particularly one study sites (Yingjiang) with lower frequency in 2012 (35%) and subsequently it raised to 90% in 2013 and 2014. Please discuss the possible cause of the spread particularly in one site. If possible, the corelate the epidemiological data of the study site for recombination point of view.

Response: Have revised according to reviewer’s advice. The high prevalence happened alongside a civil war in 2013 in north Myanmar. However, why it was F446I, but not C580Y or the other K13 genotype be the protagonist during this period was yet unknown, and no other hypothesis could be concluded based on our study. 

2.2.6 What were the QA/QC procedures for K13 sequencing? Were the sequences of the F446I containing samples confirmed; either by re-sequencing, or any other method?

Response: Firstly, our result agreed with others which sample were collected nearly in same sites and sites. Secondly, each mutant site was recheck in the sequencing figures to ensure each peak plot was single. If double, compare upstream and downstream 10bp around the site with wild type to determine it was a real mutation or misreading codon.

---

## [Decision Letter · Decision Letter 1]

16 Aug 2023

PONE-D-23-03162R1Genetic characteristics of P. falciparum parasites collected from 2012 to 2016 and anti-malaria resistance along the China-Myanmar borderPLOS ONE

Dear Dr. Li,

Thank you for submitting your manuscript to PLOS ONE. After careful consideration, we feel that it has merit but does not fully meet PLOS ONE’s publication criteria as it currently stands. Therefore, we invite you to submit a revised version of the manuscript that addresses the points raised during the review process.

Please submit your revised manuscript by Sep 29 2023 11:59PM.  If you will need more time than this to complete your revisions, please reply to this message or contact the journal office at plosone@plos.org. Please include the following items when submitting your revised manuscript:A rebuttal letter that responds to each point raised by the academic editor and reviewer(s). You should upload this letter as a separate file labeled 'Response to Reviewers'.A marked-up copy of your manuscript that highlights changes made to the original version. You should upload this as a separate file labeled 'Revised Manuscript with Track Changes'.An unmarked version of your revised paper without tracked changes. You should upload this as a separate file labeled 'Manuscript'.If applicable, we recommend that you deposit your laboratory protocols in protocols.io to enhance the reproducibility of your results. Protocols.io assigns your protocol its own identifier (DOI) so that it can be cited independently in the future. For instructions see: https://journals.plos.org/plosone/s/submission-guidelines#loc-laboratory-protocols. Additionally, PLOS ONE offers an option for publishing peer-reviewed Lab Protocol articles, which describe protocols hosted on protocols.io. Read more information on sharing protocols at https://plos.org/protocols?utm_medium=editorial-email&utm_source=authorletters&utm_campaign=protocols.

We look forward to receiving your revised manuscript.

Kind regards,

Himanshu Gupta

Academic Editor

PLOS ONE

Journal Requirements:

Reviewers' comments:

Reviewer's Responses to Questions

**Comments to the Author**

1. If the authors have adequately addressed your comments raised in a previous round of review and you feel that this manuscript is now acceptable for publication, you may indicate that here to bypass the “Comments to the Author” section, enter your conflict of interest statement in the “Confidential to Editor” section, and submit your "Accept" recommendation.

Reviewer #1: All comments have been addressed

Reviewer #2: All comments have been addressed

2. Is the manuscript technically sound, and do the data support the conclusions?

Reviewer #1: Yes

Reviewer #2: Yes

3. Has the statistical analysis been performed appropriately and rigorously? 

Reviewer #1: Yes

Reviewer #2: Yes

4. Have the authors made all data underlying the findings in their manuscript fully available?

Reviewer #1: (No Response)

Reviewer #2: Yes

5. Is the manuscript presented in an intelligible fashion and written in standard English?

Reviewer #1: (No Response)

Reviewer #2: Yes

6. Review Comments to the Author

Reviewer #1: Overall, I find the paper much improved and am satisfied with the changes. I have a few minor points though:

Introduction & Discussion – The wording needs to be checked by a dedicated editor for grammar and phrasing. I have included grammatical and phrasing suggestions where I could but I probably did not catch all them all.

Line 206-225: There is no reason to have all the statistics (the 25% per = 2.74, per = 5.02, etc etc) reported in the main text—They are all already Table 2. I suggest using the main text to highlight only the key results.

Minor comments

• Line 39: Missing a the: In (the) NW (north-west Yunnan…)

• Line 47: Change phrasing: no difference in PCT50s were present between…

• Line 47-50: having the actual values of He for each of the groups could be useful

• Line 52: would change the word inferred to hypothesized—I am more used to being used in the statistical sense (ie for model selection or statistical parameter inference)

• Line 63: backward economic development – would advise a rephrase to avoid offending people

• Line 65: Malaria is one of <the> infectious diseases – missing a the

• Line 67: “last reported local case” instead of last report oflocal <sic> case

• Line 68: Most of them (78.02, 4099/5254) were missing –missing a were

• Line 70: The emergence and spread

• Line 74: suggest replacing “because” with “following”

• Line 77-80: Rephrase as: In China, ACTs were used as the official first-line drugs to treat uncomplicated P. falciparum malaria starting in 2006.

• Line 84: Rephrase: Treatment with DHA-PIP initially demonstrated prolonged parasite clearance time( PCT), but began to fail as the partner drugs began failing in the GMS around 2010.

• Line 171: replace treated with examined

• Line 306: Rephrase “ so surveillalnce sites are usually set in these Prefectures in those days.” The phrasing is too casual for a scientific paper in my opinion

• Line 401: oppression is not the right word—advise selective pressure</sic></the>

Reviewer #2: Author have addressed all the concern and revised the manuscript accordingly. The revised version of the manuscript may accept for the publications.

7. PLOS authors have the option to publish the peer review history of their article (what does this mean?). If published, this will include your full peer review and any attached files.

Reviewer #1: No

Reviewer #2: No

---

## [Author Response · Author response to Decision Letter 1]

2 Oct 2023

I have revised the manuscript according to reviewers and editors' suggestions. The details are in the file "Response to Reviewers"

---

## [Decision Letter · Decision Letter 2]

17 Oct 2023

Genetic characteristics of P. falciparum parasites collected from 2012 to 2016 and anti-malaria resistance along the China-Myanmar border

PONE-D-23-03162R2

Dear Dr. Li,

We’re pleased to inform you that your manuscript has been judged scientifically suitable for publication and will be formally accepted for publication once it meets all outstanding technical requirements, as well as improves the English throughout the manuscript, as pointed out by the reviewer.

Kind regards,

Himanshu Gupta

Academic Editor

PLOS ONE

Additional Editor Comments (optional): There is a requirement to improve the English throughout the manuscript.

Reviewers' comments:

Reviewer's Responses to Questions

**Comments to the Author**

1. If the authors have adequately addressed your comments raised in a previous round of review and you feel that this manuscript is now acceptable for publication, you may indicate that here to bypass the “Comments to the Author” section, enter your conflict of interest statement in the “Confidential to Editor” section, and submit your "Accept" recommendation.

Reviewer #1: All comments have been addressed

2. Is the manuscript technically sound, and do the data support the conclusions?

Reviewer #1: Yes

3. Has the statistical analysis been performed appropriately and rigorously? 

Reviewer #1: Yes

4. Have the authors made all data underlying the findings in their manuscript fully available?

Reviewer #1: (No Response)

5. Is the manuscript presented in an intelligible fashion and written in standard English?

Reviewer #1: (No Response)

6. Review Comments to the Author

Reviewer #1: Line 323: “following with [the] civil war in Kachin” – remove “the”

Line 324: “We [preferred] the F446I…” -- “preferred” is the wrong word

Line 351: the high prevalence of mutant K13 [which] was …” – remove “which”

Line 356: “[Besides], this study was not planned…” replace besides with However

Line 363 “resistance to [multiply]” – replace multiply with multiple

Line 386 multiply should be replaced with multiple

Line 389 “which perhaps were interfered by PIP…” is awkward, suggest a rephrase

Line 391 remove “ayway”

Line 429-431 – “Otherwise the values of copy number…” is awkward, suggest a rephrase

7. PLOS authors have the option to publish the peer review history of their article (what does this mean?). If published, this will include your full peer review and any attached files.

Reviewer #1: No

---

## [Editor Report · Acceptance letter]

2 Nov 2023

PONE-D-23-03162R2 

Genetic characteristics of *P. falciparum* parasites collected from 2012 to 2016 and anti-malaria resistance along the China-Myanmar border 

Dear Dr. Li:

I'm pleased to inform you that your manuscript has been deemed suitable for publication in PLOS ONE. Congratulations! Your manuscript is now with our production department. 

Kind regards, 

on behalf of

Dr. Himanshu Gupta 

Academic Editor

PLOS ONE